# Perovskite Downconverters for Efficient, Excellent Color-Rendering, and Circadian Solid-State Lighting

**DOI:** 10.3390/nano9020176

**Published:** 2019-02-01

**Authors:** Ziqian He, Caicai Zhang, Hao Chen, Yajie Dong, Shin-Tson Wu

**Affiliations:** 1College of Optics and Photonics, University of Central Florida, Orlando, FL 32816, USA; zhe@knights.ucf.edu (Z.H.); hao.chen@knights.ucf.edu (H.C.); Yajie.Dong@ucf.edu (Y.D.); 2Department of Materials Science & Engineering, University of Central Florida, Orlando, FL 32816, USA; cczhang@knights.ucf.edu; 3NanoScience Technology Center, University of Central Florida, Orlando, FL 32826, USA

**Keywords:** perovskite nanocrystals, powders, solid-state lighting, optimization, circadian effect

## Abstract

Advances in materials, color rendering metrics and studies on biological effects promote the design for novel solid-state lighting sources that are highly energy efficient, excellent at color rendering and healthy for human circadian rhythms. Recently, perovskite nanocrystals have emerged as narrow-band, low-cost, color-tunable downconverters, elevating the design and development of solid-state lighting to a new level. Here, we perform a systematic optimization of using perovskite nanocrystals as downconverters to simultaneously optimize vision energy efficiency, color rendering quality and circadian action effect of lighting sources at both fixed and tunable color temperatures. Further analysis reveals the inherent differences in central wavelength and bandwidth preferences for different cases, providing a general guideline for designing circadian lighting. Through systematic optimization, highly efficient circadian lighting sources with excellent color rendering can be achieved.

## 1. Introduction

During the past few decades, solid-state lighting (SSL) based on white light-emitting diodes (LEDs) has been gradually overtaking incandescent and fluorescent lighting sources due to its lower power consumption, longer lifetime, smaller size, higher brightness and faster response time [1]. While much effort has been devoted to improving its efficiency [2], with researchers of SSL aiming beyond energy saving to applications such as circadian lighting or horticultural lighting, tailoring the spectra in terms of specific requirements has become increasingly important [3]. Moreover, the color quality metrics of SSL have advanced [4]. To overcome the deficits of the outdated color rendering index (CRI), many proposals have been suggested, such as color quality scale (CQS), color fidelity index (CFI), and color gamut index (CGI). For example, CQS is calculated using 15 Munsell test color samples and is more suitable for artificial light sources that enhance the object chroma [5]. In parallel to the advances of color quality metrics, the biological impact of light has also been studied extensively [6,7,8]. Under ambient light, the intrinsically photosensitive retinal ganglion cells (ipRGCs) innervating the suprachiasmatic nucleus (SCN) can influence the melatonin secretion, and thus affect the circadian rhythm [9]. In reference to the experiments on light-induced melatonin suppression, the action spectrum of the circadian effect with a peak at the blue wavelength has been proposed [10,11]. It is indeed necessary to take the circadian effect into design consideration for achieving circadian lighting [12,13,14,15,16].

While the development of high-performance blue LEDs [17] has enabled widespread applications of phosphor-converted white LEDs, the rapid growth of new narrow-band phosphors and quantum dots has advanced the study of white LEDs further, in that narrow-band downconverters are believed to provide better vision performance [18,19]. Recently, metal halide perovskite (MHP) nanocrystals have emerged as novel, narrow-band, low-cost, and color-tunable downconverters [20,21,22,23,24,25], promoting the design and development of liquid crystal display (LCD) backlighting [26] and SSL [12,27] to a new level. However, the use of MHP nanocrystals in practical applications needs to overcome the instability issue under external stresses [28,29]. One promising solution is to synthesize these MHP nanocrystals in or on a polymer or inorganic matrix [30,31]. More interestingly, our group has developed stable, scattering, downconverting particles through anchoring the MHP nanocrystals on light-diffusing polymer particle matrices [32]. The simple process and very low-cost precursors and materials should enable their adoptions for these applications.

In this paper, we perform numerical calculations to systematically optimize the performance of white SSL sources, aiming both at fixed correlated color temperatures (CCTs) and tunable CCTs. Using MHP nanocrystals as downconverters, vision energy efficiency, color rendering quality and the circadian effect of the light sources can be simultaneously optimized. Note that previous works [12,27] fabricated MHP nanocrystals and mixed their spectra to demonstrate tunable SSL sources with high performance. However, the MHP nanocrystals they fabricated were not stable enough and their performance was not theoretically optimized. Here, we fabricate MHP-polymer composite powders and films that emit different colors and take their optical properties into design considerations. Through our method, the great potential of using MHP as downconverters for advanced SSL can be explored. Meanwhile, the precedents in lighting optimizations [13,15,16] have focused on two-objective optimizations. Instead, we apply three-objective optimizations to exploit the ultimate tradeoffs between the objectives and present the correlations between the input parameters and the objectives, which are meaningful for further SSL designs. Through the proposed optimization method, highly efficient circadian SSL sources with excellent color rendering are achievable.

## 2. Materials and Methods

### 2.1. Perovskite-Polymer Composite Powders

MHP-polymer composite powders have recently emerged as downconverters with narrow bandwidth, good stability against external stresses, high photoluminescence (PL) efficiency and low cost. Using commercial light-diffusing polymer powders as a matrix, the synthesized composite powders not only provide excellent downconverting abilities, but also offer outstanding light-diffusing capabilities. More intriguingly, the emission central wavelengths can vary with respect to different cations or halide compositions, and the entire visible band can be covered. The full width at half maximum (FWHM) of each emitter can also be adjusted according to many factors, such as differences in cations and ligands. Due to the abovementioned advantages, MHP can be a good candidate for SSL and LCD backlights.

To fabricate the MHP-polymer composite powders, CsX (X = Br, I) and PbX_2_ precursor with a 1:1 molar ratio was first dissolved in *N,N*-dimethylformamide (DMF), while CsCl and PbCl_2_ precursor was dissolved in dimethyl sulfoxide (DMSO), to obtain an overall concentration of 0.04 mmol/mL. Oleylamine (OAm) and oleic acid (OA) were also mixed into the precursor solution with concentrations of 10 μL/mL and 100 μL/mL, respectively, except for CsPbI_3_, where only OAm was applied as surfactant. To get different compositions of MHP (listed in Table 1), we mixed the precursors according to their stoichiometric ratios. Then, commercial light diffusing powders made of either polystyrene (PS) or polymethylmethacrylate (PMMA) were mixed with the precursor solution at a concentration of 500 mg/mL. To induce the formation of MHP-polymer composite powders, antisolvent isopropyl alcohol (IPA) was quickly dropped into the stirring dispersion within 5 s. After centrifuging (4000 rpm) for several minutes, the precipitate was finally vacuum-dried overnight to obtain the powders.

These powders can be further molded into different form factors. As an example, the composite powders can be dispersed into polymer precursors to form diffusing downconverting films. Specifically, the composite powders were added into the silicone elastomer base (SYLGARD 184 Silicone Elastomer Kit) with a weight ratio of 1:20 (composite powders : silicone elastomer base) and sonicated for 1 h to fully disperse the powders. Then, the curing agent with a volume ratio 1:7 (curing agent : base set) was dropped into the base and the mixture was stirred for 5 min. After vacuum-drying overnight and baking at 80 °C for 4 h, homogeneous downconverting films were achieved.

Figure 1 demonstrates some fabricated powders and films where vivid colors can be observed under 365-nm UV light excitation. Through altering the anion compositions, the whole visible band can be covered. We measured the PL quantum efficiency (PLQY) of green and red samples in a FluoroMax Plus (Horiba Jobin Yvon, Kyoto, Japan) spectrofluorometer equipped with the Quanta-phi integrating sphere. The green powders exhibit about 70% PLQY, and the red ones achieve slightly lower PLQY (~40%). Since in our configuration a blue LED is applied to pump the MHP downconverters, detailed characterization of their photoluminescent properties (measured by Ocean Optics Spectrometer USB 2000+ (Ocean Optics, Largo, FL, USA)) under a blue light excitation is needed, as summarized in Table 1. It can be noted that for cyan and green downconverters, the FWHMs are narrow, while the yellow and red downconverters are relatively broad. To utilize these stable downconverters, variations of the FWHMs in relation to the emission peaks needs to be considered in the optimization.

### 2.2. Systematic Optimization

Figure 2 depicts two strategies of adopting MHP-polymer composite powders into SSL. One is to incorporate several powders with different emission wavelengths into a single film and utilize a blue LED as pumping beam. The other is a multi-package approach where each package emits a single color, which can be either a LED or a blue LED-pumped single-color powder-enveloped film. The first strategy is of lower cost and simpler configuration, while the second allows wide CCT tunability by simply modifying the intensity ratio among them.

To evaluate the first approach, one blue LED and five MHP downconverters are involved to generate white light. Here, more than four spectra are chosen to explore the trade-offs between the objectives thoroughly. As mentioned, each downconverter can emit light with a different central wavelength and FWHM. According to the experiments, the blue-green downconverters usually have a relatively narrow FWHM and the yellow-red downconverters show a broader FWHM. The central wavelength of the blue LED is also allowed to change in a small range but with fixed FWHM, as often seen in commercial products. Detailed parameters are listed in Table 2. As Table 2 illustrates, the MHP downconverters are divided into different bands, in accordance with their typical FWHMs. The split is also helpful for characterizing the correlation between each band and each optimization objective, which will be discussed later. It is noticed that some overlaps between different bands exist, in order to provide more degree of freedom in the design. Once the central wavelengths and FWHMs are determined, we can approximate the spectral power distribution (SPD) of each emitter using Gaussian function [33]:(1)$$S_{i}(\lambda ,\lambda_{0},\Delta \lambda )=e^{-4\ln2\frac{\left(\lambda -\lambda_{0}\right)}{\Delta \lambda^{2}}}$$
where *i* stands for the blue LED and MHP 1 to 5, *λ*_0_ is the central wavelength, and Δ*λ* is the FWHM of the corresponding emission spectrum. Subsequently, the tristimulus values (*X_i_*, *Y_i_*, *Z_i_*) of each spectrum can be calculated using the color matching function. Under a specified CCT, the tristimulus values (*X_w_*, *Y_w_*, *Z_w_*) of the reference white light source can also be determined. Then, the six spectra can be mixed to generate a white light with the same tristimulus values through:(2)$$\left[\begin{array}{ccc} X_{1} & \cdots & X_{6}\\ Y_{1} & \cdots & Y_{6}\\ Z_{1} & \cdots & Z_{6} \end{array}\right]\left[\begin{array}{c} a_{1}\\ \vdots \\ a_{6} \end{array}\right]=\;\left[\begin{array}{c} X_{w}\\ Y_{w}\\ Z_{w} \end{array}\right]$$
where *a*_1_ to *a*_6_ stand for the ratio of each spectrum. Since Equation (2) can eliminate 3 dependent variables, there will be 14 input variables in the optimization, including 6 central wavelengths, 5 FWHMs and 3 spectrum ratios.

As mentioned, a good SSL source should have high vision energy efficiency, excellent color rendering quality and appropriate circadian effect according to the CCT. For this purpose, 3 objectives are chosen, namely, luminous efficacy of radiation (LER), CQS and circadian action factor (CAF). LER reflects the efficiency of the output light that can be converted to brightness received by human eyes, which is defined by:(3)$$LER=\frac{683\frac{lm}{W}\int S_{tot}\left(\lambda \right)V\left(\lambda \right)d\lambda }{\int S_{tot}\left(\lambda \right)d\lambda }$$
where *S_tot_*(*λ*) is the SPD of the mixed white light and *V*(*λ*) is the photopic eye sensitivity function. For color quality metric, instead of applying CRI as in our previous work [34], CQS is employed since it is much more suitable for narrow-band emitters and it outperforms in many other aspects [5]. CAF characterizes the circadian effect of light, which can be quantified by:(4)$$CAF=\frac{K_{c}\int S_{tot}\left(\lambda \right)C\left(\lambda \right)d\lambda }{\int S_{tot}\left(\lambda \right)V\left(\lambda \right)d\lambda }$$
where *C*(*λ*) is the circadian action function [10] and *K*_c_ is a normalization constant which ensures CAF = 1 for the International Commission on Illumination (CIE) standard daylight illuminant D65.

After defining all the variables and objectives, we then optimize the performance for two white light sources as examples: one at low CCT (2700 K) and one at high CCT (6500 K). While LER and CQS need to be maximized for both cases, CAF at 2700 K should be as low as possible and at 6500 K should be as high as possible to follow the circadian phase. To achieve the global optimal solutions, four optimization algorithms, namely, the genetic algorithm (GA), particle swarm optimization (PSA), differential evolution (DE) and adaptive simulated annealing (ASA), are applied interchangeably during the optimization. For general multi-objective optimizations, there will be tradeoffs between the objectives, resulting in a set of optimal solutions where each individual will have at least one objective outperforming that of others. Such a set of optimal solutions creates a Pareto front [35]. In the optimization, more than 10,000,000 iterations were performed to obtain more than 2000 optimal solutions for each case.

For the multi-package approach aiming at tunable SSL sources, since more packages will require more sophisticated driving circuits and color-mixing optics, fewer packages are generally preferred. In our calculations, 4 packages, namely, a blue LED, a green downconverting film, a yellow downconverting film and a red LED (RYGB) are involved. This choice makes use of high-efficiency LEDs and avoids the green gap. The parameter ranges of each emitter are almost the same as those in the first approach except that the FWHM of the red LED is fixed at 20 nm. Specifically, the green downconverting film is a combination of MHP1 and MHP2 while the yellow film is a combination of MHP3 and MHP4. The study focuses on exploring the tradeoffs among maximum CAF tunability from a low CCT (2700 K) to a high CCT (6500 K), average LER, and average CQS of these two CCTs. The objective functions are defined as:(5)$$vCAF=CAF_{6500K}/CAF_{2700K}$$
(6)$$aLER=\left(LER_{2700K}+LER_{6500K}\right)/2$$
(7)$$aCQS=\sqrt{\left(CQS_{2700K}^{2}+CQS_{6500K}^{2}\right)/2}$$

It is worth mentioning that the average CQS is calculated using the root mean square to avoid large discrepancies between the two cases. In total, there will be 8 input variables including 4 central wavelengths, 2 FWHMs and 2 spectrum ratios. The abovementioned optimization method is then applied to calculated and investigate the Pareto front.

## 3. Results and Discussion

### 3.1. Fixed CCTs

Figure 3a illustrates the Pareto front of the three-objective optimization for a 2700-K SSL source. Each point on the Pareto front surface is an optimal solution. In the low CCT case, an SSL source with low CAF will sacrifice CQS and LER. The surface geometry depicts the intrinsic tradeoffs between vision energy efficiency, color rendering quality and circadian effect. It can be noticed that CAF and CQS are strongly exclusive, while LER is only severely restricted by the other two objectives at very high values. Within the set of optimal solutions, special optimal solutions can be selected in terms of application needs. Here one example, optimal solution 1 (OP1), with high CQS, low CAF and high LER, is demonstrated as the red dot in Figure 3a. The parameters of OP1 are listed in Table 3 and the corresponding SPD is plotted in Figure 3b. It offers 90.1 CQS, 0.33 CAF and 372.4 lm/W LER, while CRI = 92.8 is also given as a reference. In comparison with the reference white, which is a blackbody radiator at 2700 K, OP1 has much higher LER and lower CAF, while keeping CQS at a relatively high level. More interestingly, the SPD of OP1 only exhibits four peaks, indicating that six spectra are excessive. To validate, the SPD is fitted with four Gaussian functions. The parameters and the SPD of the fitted result are shown in Table 3 and Figure 3b, respectively. It turns out that the performance of the fitted result is very similar to that of OP1.

On the other hand, Figure 3c illustrates the Pareto front for a 6500-K SSL source. In this case, an SSL source with high CAF will sacrifice CQS and LER. Similar to the previous case, CAF and CQS are strongly exclusive, while LER is only severely restricted by the other two objectives at a very high value. Again, we choose an optimal solution to study in detail. Here, OP2 is selected as an ultra-high CQS, moderate CAF (comparable to reference white D65) and high LER source, highlighted as the red dot in Figure 3c. The parameters of OP2 are summarized in Table 4 and the corresponding SPD is depicted in Figure 3d. It provides 95.3 CQS, 1.0 CAF and 306.8 lm/W LER, while CRI = 95.0 is also given as a reference. Compared to the reference white, CIE standard illuminant D65 (6504 K), OP2 has higher LER and the same CAF, while maintaining CQS at a high level. Similar to the low CCT case, six spectra are excessive in that the SPD of OP2 exhibits only five peaks. This SPD is then fitted with five Gaussian functions. The parameters and the SPD of the fitted result are illustrated in Table 4 and Figure 3d, respectively. It turns out that the fitted result performs almost the same as OP2.

An advantage of the systematic optimization is that the parameter preference as well as the correlations between each input parameter and each objective can be studied. Figure 4a–f plots the relationship between the objectives and the central wavelengths and Figure 4g–l depicts the relationship between the objectives and the bandwidths, respectively, for the low CCT case. The central wavelength of MHP5 tends to be negatively correlated to LER while that of others are almost irrelevant to LER. Meanwhile, the central wavelengths of the blue LED and MHP2 have higher impact on CQS and CAF than those of others. Interestingly, the behaviors of CQS and CAF are very similar, proving a strong exclusion between these two objectives. On the other hand, the bandwidths of the spectra have almost no impact on all three objectives. Noticeably, however, the bandwidth of MHP5 prefers to be narrow for all the optimal solutions. Meanwhile, the correlation analysis for the high CCT case is concluded in Figure 5. Unlike the 2700-K case, the central wavelengths of the blue LED, MHP1, MHP2 and MHP3 are all confined in small ranges. It also can be seen that for high LER, the red light tends to have a shorter central wavelength. However, the relationship between the objectives and the bandwidths is similar to the previous case where they are not correlated except that this time the bandwidth of the MHP1 inclines to be narrow.

From the above correlation analysis, there are intrinsic differences in central wavelength and bandwidth preferences between high CCT and low CCT. The study is beneficial for designing SSL sources at different CCTs as well as CCT-tunable SSL sources in terms of different requirements. Another interesting aspect is that in our configuration, six narrow-band spectra are already excessive for both cases. It indicates that for the multi-package approach, there is no need of combining as many emitters as possible under the metrics applied in our case. For such an approach, extra optics for color mixing and sophisticated driving circuits are usually needed. By choosing an appropriate number of emitters, the design can be optimized for both SSL performances and design complexities.

### 3.2. Tunable CCTs

Figure 6a depicts the Pareto front of the three-objective optimization for the multi-package approach. The three objectives, vCAF, aLER and aCQS, are strongly mutually exclusive, as can be seen from the triangular-like geometry. This means that an increment of one objective will sacrifice the other objectives severely. A special point is picked out for detailed studies (OP3), denoted as the red point in Figure 6a. OP3 has 3.52 vCAF, 348.0 lm/W aLER and 86.2 aCQS. Specifically, it provides 0.29 CAF, 390.5 lm/W LER and 80.3 CQS at 2700 K and 1.02 CAF, 305.6 lm/W LER and 91.7 CQS at 6500 K. We can further implement the intermediate CCTs to ensure the tunability in between these two CCTs. Since there are more than three colors, we can always choose points that lie on the blackbody locus and finetune the performance of the intermediate CCTs. An example of SPD variation is shown in Figure 6b. The details of the CAF, LER, CQS, CRI, R9 (deep red) and R13 (skin tone) values as a function of CCTs are further demonstrated in Figure 6c. The optimized solution performs very well in rendering deep red and skin tone colors. Moreover, it can follow the circadian phase and offer lower and higher CAF than corresponding standard illuminants at 2700 K and 6500 K, respectively. Thus, the example is a perfect SSL source with large CAF tunability, high efficiency and excellent color rendering.

Correlation analysis is also performed, and results are depicted in Figure 7. For higher average LER, the red LED inclines to have shorter central wavelengths, while the blue LED and the green downconverting film tend to have longer central wavelengths. In the meantime, higher average CQS requires longer central wavelengths of the red LED and the yellow downconverting film. Interestingly, large CAF variability confines all four central wavelengths. This can be a guideline for designs of large CAF-tunability four-hump SSL sources. On the other hand, the bandwidths of the two downconverting films seem not strongly correlated to average LER and CQS. However, for large CAF tunability, the yellow film tends to have broader bandwidths while the green film prefers narrower bandwidths.

## 4. Conclusions

We fabricated perovskite-polymer composite powders and demonstrated downconverting films which incorporated composite powders with polymer matrices. Based on the optical properties of the composite powders, we further performed numerical calculations to systematically optimize the performance of two kinds of white SSL sources: one with fixed CCTs (2700 K and 6500 K) and the other with tunable CCTs. Using MHP nanocrystals as downconverters, vision energy efficiency, color rendering quality and the circadian effect of the light sources can be simultaneously optimized. Further correlation analysis revealed the intrinsic differences in central wavelength and bandwidth preferences for different cases. The proposed optimization method not only provides optimal solutions for highly efficient circadian SSL sources with excellent color rendering, but also offers a general guideline for improving SSL performance.

## Figures and Tables

**Figure 1 nanomaterials-09-00176-f001:**
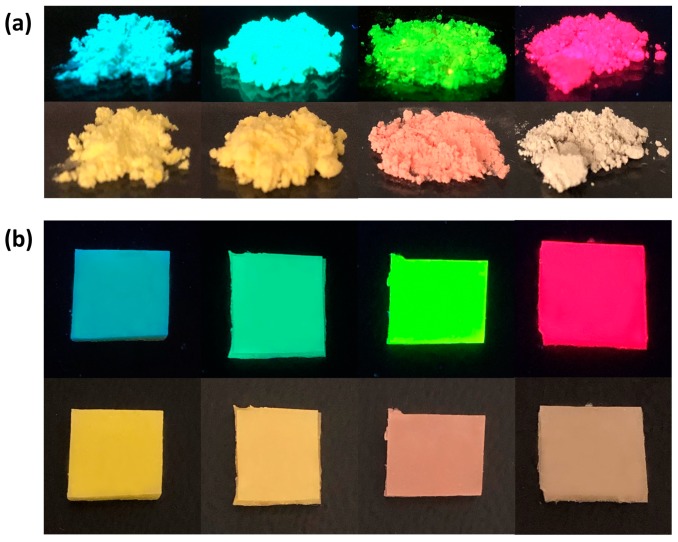
Pictures of fabricated (**a**) metal halide perovskite (MHP)-polymer composite powders and (**b**) downconverting films with different anion compositions. The upper row is under 365-nm UV light excitation and the lower row is under ambient light.

**Figure 2 nanomaterials-09-00176-f002:**
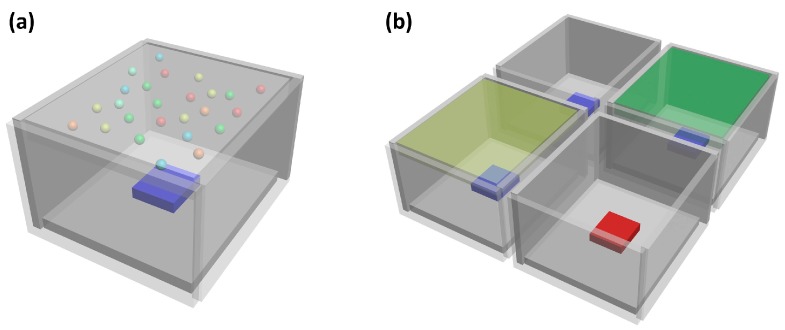
Two strategies of adopting MHP into solid-state lighting (SSL): (**a**) powders-in-film and (**b**) multi-package.

**Figure 3 nanomaterials-09-00176-f003:**
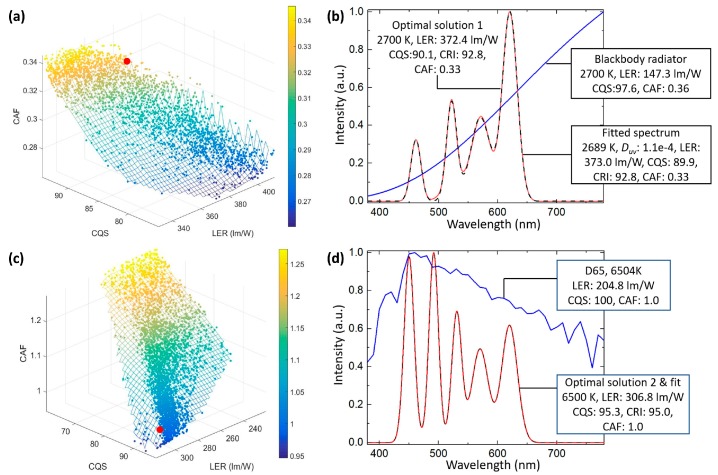
The Pareto front of the 3-objective optimization for (**a**) a low correlated color temperature (CCT; 2700 K) and (**c**) a high CCT (6500 K) SSL source. The red dots denote (**a**) optimal solution 1 (OP1) and (**c**) optimal solution 2 (OP2). (**b,d**) The spectral power distribution (SPD) and performance of the reference whites (blue solid line), OPs (red solid line) and the fitted results (black dashed lines) for (**b**) 2700-K and (**d**) 6500-K CCTs.

**Figure 4 nanomaterials-09-00176-f004:**
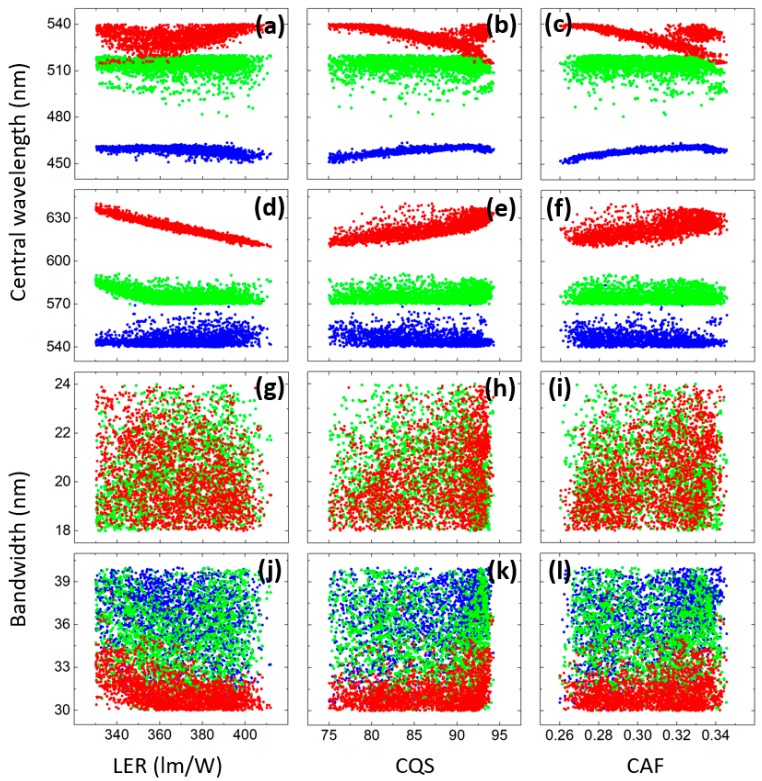
(**a**–**f**) The correlations between the objectives and the central wavelengths of (**a**–**c**) blue LED (blue), MHP1 (green), MHP2 (red), (**d**–**f**) MHP3 (blue), MHP4 (green), MHP5 (red); and (**g**–**l**) the correlations between the objectives and the bandwidths of (**g**–**i**) MHP1 (green), MHP2 (red), (**j**–**l**) MHP3 (blue), MHP4 (green), MHP5 (red) at 2700-K CCT.

**Figure 5 nanomaterials-09-00176-f005:**
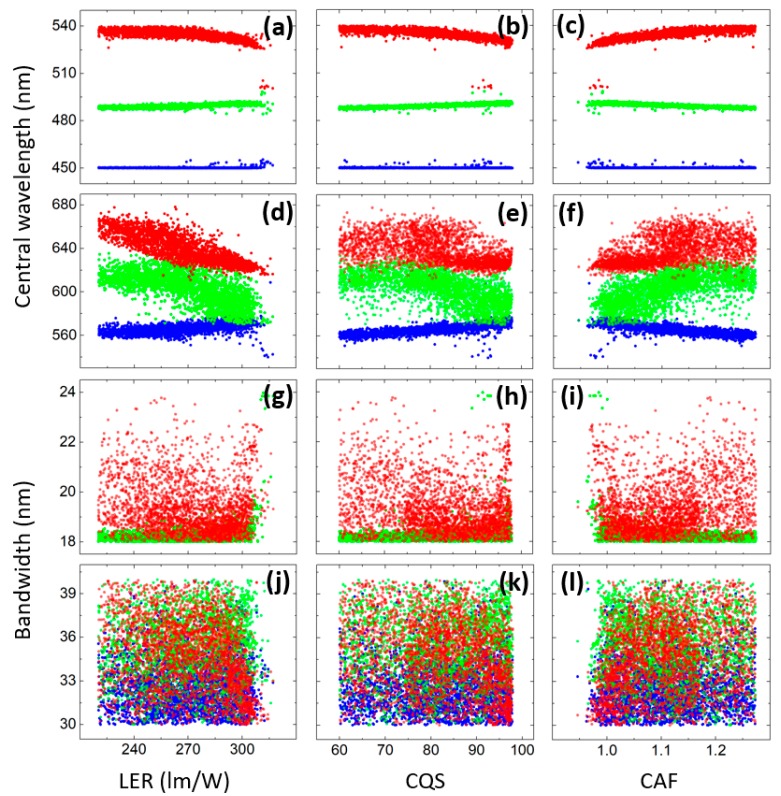
(**a**–**f**) The correlations between the objectives and the central wavelengths of (**a**–**c**) blue LED (blue), MHP1 (green), MHP2 (red), (**d**–**f**) MHP3 (blue), MHP4 (green), MHP5 (red); and (**g**–**l**) the correlations between the objectives and the bandwidths of (**g**–**i**) MHP1 (green), MHP2 (red), (**j**–**l**) MHP3 (blue), MHP4 (green), MHP5 (red) at 6500-K CCT.

**Figure 6 nanomaterials-09-00176-f006:**
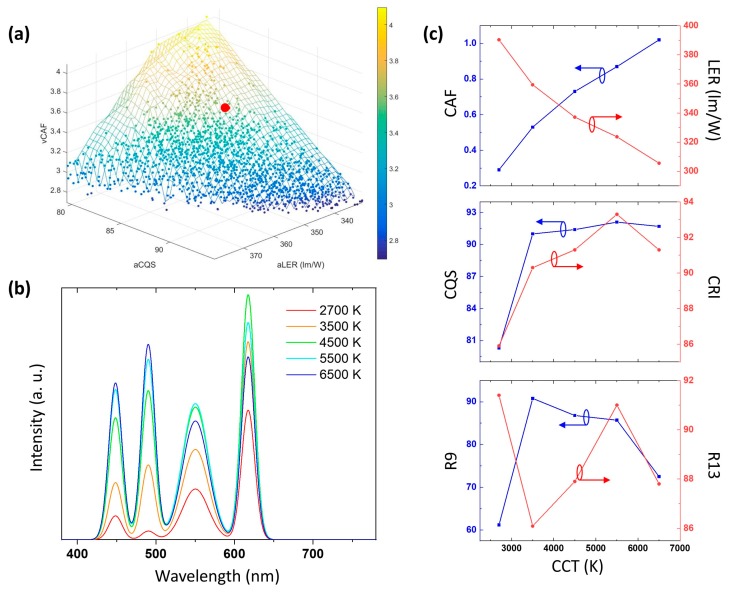
(**a**) The Pareto front of the three-objective optimization for a tunable SSL source where the red dot denotes optimal solution 3 (OP3). (**b**) SPD at variable CCTs for an example. The peak wavelengths of the components are 448.5 nm, 490.3 nm, 550.3 nm, and 617.3 nm. The bandwidths are 19.9 nm and 39.1 nm for green and yellow, respectively. (**c**) Details of the CAF, LER, CQS, CRI, R9 and R13 values as a function of CCTs.

**Figure 7 nanomaterials-09-00176-f007:**
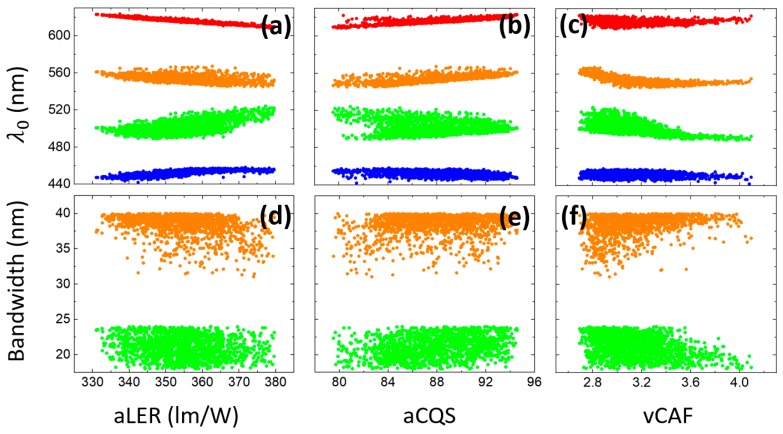
(**a**–**c**) The correlations between the objectives and the central wavelengths of blue LED (blue), green downconverting film (green), yellow downconverting film (orange), and red LED (red); (**d**–**f**) the correlations between the objectives and the bandwidths of green downconverting film (green) and yellow downconverting film (orange) for CCT-tunable SSL sources.

**Table 1 nanomaterials-09-00176-t001:** Emission peaks, full width at half maximums (FWHMs) and corresponding International Commission on Illumination (CIE) 1931 color coordinates of MHP-polymer composite powders under a 450-nm blue light-emitting diode (LED) excitation.

MHP Components	Emission Peaks (nm)	FWHMs (nm)	CIE 1931 Color Coordinates (x, y)
CsPb (Cl_0.33_Br_0.67_)_3_	483.6	20.7	(0.084, 0.177)
CsPb (Cl_0.25_Br_0.75_)_3_	499.6	18.5	(0.024, 0.514)
CsPbBr_3_	526.0	18.0	(0.132, 0.806)
CsPb (Br_0.5_I_0.5_)_3_	558.2	34.2	(0.372, 0.622)
CsPb (Br_0.33_I_0.67_)_3_	609.3	34.2	(0.642, 0.358)
CsPb (Br_0.25_I_0.75_)_3_	626.0	37.0	(0.682, 0.318)
CsPb (Br_0.2_I_0.8_)_3_	634.2	39.5	(0.693, 0.306)
CsPb (Br_0.14_I_0.86_)_3_	664.0	31.5	(0.726, 0.274)

**Table 2 nanomaterials-09-00176-t002:** Parameter ranges of each spectrum.

	Blue LED	MHP1	MHP2	MHP3	MHP4	MHP5
Central wavelength (nm)	450–465	480–520	500–540	540–610	570–640	610–680
FWHM (nm)	20 (fixed)	18–24	18–24	30–40	30–40	30–40

**Table 3 nanomaterials-09-00176-t003:** Parameters of OP1 and the fitted result.

		Blue LED	MHP1	MHP2	MHP3	MHP4	MHP5
	Central wavelength (nm)	461.4	502.4	521.7	540.4	572.4	620.7
OP1	FWHM (nm)	20.0	23.2	19.0	35.5	35.3	30.0
	Ratio (%)	13.8	1.0	20.9	3.4	18.6	42.3
	Central wavelength (nm)	461.4	522.0	570.9	620.8	-	-
Fitting	FWHM (nm)	20.1	21.0	39.5	29.7	-	-
	Ratio (%)	14.2	22.8	19.4	43.6	-	-

**Table 4 nanomaterials-09-00176-t004:** Parameters of OP2 and the fitted result.

		Blue LED	MHP1	MHP2	MHP3	MHP4	MHP5
	Central wavelength (nm)	450.3	491.9	530.8	570.0	571.6	620.1
OP1	FWHM (nm)	20.0	18.7	19.9	32.9	36.3	31.6
	Ratio (%)	25.9	26.5	18.1	11.4	1.7	16.4
	Central wavelength (nm)	450.3	491.9	530.8	570.2	620.1	-
Fitting	FWHM (nm)	20.0	18.7	19.9	33.3	31.6	-
	Ratio (%)	25.9	26.5	18.1	13.1	16.4	-

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
