# Peer review of "Perovskite Downconverters for Efficient, Excellent Color-Rendering, and Circadian Solid-State Lighting"

_nanomaterials, 2019, doi:10.3390/nano9020176_

Reviewer 1 Report

In this manuscript, the authors present systematic numerical calculations regarding the optimization of the performance of white solid-state lighting sources at both fixed and tunable correlated color temperatures using polymer-Metal halide nanocrystals composites, that they have designed and fabricated, as efficient downconverters. Their work is original and nicely organized and presented. Interesting and, at the same time, significant results are reported and quite effectively are conveyed to the reader. I was very pleased to see the systematic optimization performed on both proposed/employed strategies for adopting the polymer-perovskite nanocrystal composites as downconverters into solid-state lighting sources, solely based on their optical properties. The proposed method not only optimizes vision energy efficiency, color rendering quality and circadian rythm effect of the employed solid-state light sources but also offer more general quidelines to improve their performance. Overall, this is scientific work of high merit and deserves to be published in its present form.  

Author Response

Please see my uploaded response letter. 

Reviewer 2 Report

I  don't see the significant novelty to recommend publication of this draft for the Journal of Nanomaterials. The draft is very technical and presented numerical calculation is nothing special and can be done using different computational electromagnetics packages. In the presented form the draft is more suitable for some technical/optical journal. Additionally, different perovskite structures were already studied and described that I don't see significant novelty regarding perovskite physics in the presented study.

Author Response

Please see my uploaded response letter. 

Reviewer 3 Report

The manuscript from Z. He et al. reports on perovskite nanocrystals used as downconverter for Solid-State Lighting (SSL) sources. Development of new SSL sources with low power consumption and high demand lighting quality is becoming key issue. Thanks to their outstanding optoelectrical properties, halide perovskite nanocrystals are recently emerging as new SSL sources. In this manuscript, the authors show an optimization method based on optical properties of the perovskite composite materials and numerical calculations to improve SSL source performance.

The paper presents interesting results and is of great interest to material science community, but the paper is confusing when reading and need more scientific method. Therefore, I recommend major revisions before acceptance for publication in Nanomaterials journal. The main points are mentioned below.

1.     I do not see clearly in the paper what are experimental results and simulation results. For example, the luminous efficiencies reported in Figure 3(b, c) are obtained from sample measurements or simulation results? The distinction between experiments and simulations as well as the link between should be clearly emphasised.

2.     Experimental details on the materials preparation are missing. The preparation and composition of MPH-polymer composites reported in Table 2 should be detailed in order to be able to reproduce the results.

3.     In table 1, the authors show the emission properties of MHP-polymer composites powder excited by 450 nm blue LED but the photoluminescence quantum efficiencies should be reported for each set of material in order to judge the real efficiency of the materials.

4.     The title of the paper is not correct. The authors claimed that their new SSL sources based on perovskite nanocrystals are healthy. But the perovskite materials used contain lead and caesium, which are toxic elements to be avoided (particularly lead) in new systems for broad applications.

Author Response

Please see my uploaded response letter. 

Reviewer 4 Report

The authors have fabricated composite powders of perovskite-polymer to demonstrate the downconverting films. Numerical calculations show optimization of the performance of two types of white solid-state-light sources: one with fixed CCTs (2700 K and 6500 K) and the other one with tunable CCTs. They have used metal halide perovskite nanocrystals as downconverters and vision energy efficiency, color rendering quality and circadian effect of the light sources were optimized. The proposed method shows optimal solutions for highly efficient, excellent color rendering, healthy SSL sources, and understanding the knowledge for improving the performance of solid-state-light. 

The work is novel in nature and interesting to the readers as well the light science community. However, I feel that the work should be improved with respect to the following points:

1. The background could be provided more interestingly and precisely.

2. The experimental section could be improved to understand the results of your optimization through numerical analysis.

3. Please describe the methods of analysis sufficiently.

Author Response

Please see my uploaded response letter. 

Round  2

Reviewer 2 Report

I see significant improvement from the first version and the presented draft satisfied minimum condition in order to be accepted in the unchanged form. 

Reviewer 3 Report

The revised version can be published in the present form.